



# Has fire policy decreased the return period of the largest wildfire events in France? A Bayesian assessment based on extreme value theory

Guillaume Evin[1], Thomas Curt[2], and Nicolas Eckert[1]

[1]Irstea - UR ETGR Erosion Torrentielle, Neige et Avalanches, Université Grenoble Alpes, 38402 Saint-Martin-d'Hères, France
[2]Irstea - RECOVER Mediterranean Ecosystems and Risks, 3275 route Cézanne - 13182 Aix-en-Provence, France

**Correspondence:** Guillaume Evin (guillaume.evin@irstea.fr)

**Abstract.** Very large wildfires have high human, economic and ecological impacts so that robust evaluation of their return period is crucial. Preventing such events is a major objective of the new fire policy set up in France in 1994, which is oriented towards fast and massive fire suppression. Whereas this policy is probably efficient for reducing the mean burned area (BA), its effect on the largest fires is still unknown. In this study, we make use of statistical Extreme Value Theory (EVT) to compute

return periods of very large BA in southern France, for two distinct periods (1973 to 1994, and 1995 to 2016) and for three pyroclimatic regions characterized by specific fire activities. Bayesian inference and related predictive simulations are used to fairly evaluate related uncertainties. Results demonstrate that the BA corresponding to a return period of 5 years has actually significantly decreased, but that this is not the case for large return periods (e.g. 50 years). For example, in the most fire-prone region, which includes Corsica and Provence, the median 5-year return level decreased from 5,000 ha. to 2,400 ha., while the

median 50-year return level decreased only from 17,800 ha. to 12,500 ha. This finding is coherent with the recent occurrence of conflagrations of large and intense fires clearly far beyond the suppression capacity of firemen. These fires may belong to a new generation of fires promoted by long-term fuel accumulation, urbanization into the wildland, and ongoing climate change. These findings may help adapting the operational system of fire prevention and suppression to ongoing changes. Also, the proposed methodology may be useful for other case studies worldwide.

# 1   Introduction

Wildfires are important hazards and a major ecological disturbance. In southern France, 2500 fires are reported each year over the recent period, and burn an average of approximately 12,000 ha. (Curt and Frejaville, 2018). Burned area (BA) distribution is highly asymmetric with a large amount of small fires, and rare large ones. However, if the largest fires (> 100 ha.) represent only 1% of the total number of fires, they account approximately for 70% of the total BA and they consume two-thirds of the total

annual budget dedicated to civil protection against fire risk (Chatry et al., 2010; Curt and Frejaville, 2018). In addition, such large and destructive wildfires are likely to increase in southern Europe due to changes in climate and landscape (Bedia et al., 2014; Oliveira et al., 2014). As a consequence, establishing new tactics and strategies for a better prevention and preparedness





to face large fires has become a centerpiece of the European fire policy (EFIMED, 2011; Tedim et al., 2016). Indeed, wildfires are particular hazards because they are fought and suppressed in real time by firemen, at least in Europe where almost no fire propagates freely. Hence, the final BA depends upon the balance between environmental drivers that favor fire enlargement (remote terrain, strong wind, or connected fuels Fernandes et al., 2016) and the efficacy of suppression tactics (Lahaye et al., 2014).

For many geophysical variables and/or hazards such as rainfall, snowfall or river discharge, protective measures are designed to withstand an event with a given small exceedance probability, i.e., an event that is generally much rarer than those already observed (Sharma et al., 2012). As a consequence, a robust evaluation of the return period of these extreme events is of utmost importance to risk mitigation (Read and Vogel, 2015; Volpi et al., 2015). Concerning fires, this information helps pre-determining the size of the fire crews and of fire tactical means such as airplanes and trucks, which support ground forces during extreme fire events (Lahaye et al., 2014). It may also help governmental agencies and reinsurance companies to evaluate the cost of future fires in the current context of ongoing changes in climate and landscapes (Malamud et al., 2005; EFIMED, 2011).

Yet, to date, estimating return levels of fires corresponding to large return periods (e.g. > 10 years) is rarely done and few dedicated studies are available. For instance, in southern France, no such study exists although this region is fire-prone and comprises a vast array of human, economic and ecological assets at risk (Curt et al., 2016). In the literature, wildfire quantiles corresponding to large return periods have been fitted using power-law distributions (Malamud and Turcotte, 1999; Ricotta et al., 2001; Malamud et al., 2005). Cumming (2001) applies a truncated exponential distribution. Reed and McKelvey (2002) apply a stochastic model for the spread and extinguishment of fires, which is used to illustrate the deviation of the upper-tail from a simple power law distribution. However, as for other geophysical variables, Extreme Value Theory should be preferred due to its strong mathematical groundings (Coles, 2001). Specifically, in the univariate case, the generalized extreme value (GEV) distribution, the generalized Pareto distribution (GPD), or the combination of the GEV distribution with a Poisson point process (PP) are the representations of interest (Moritz, 1997; Alvarado et al., 1998; Holmes et al., 2008; Jiang and Zhuang, 2011; Hernandez et al., 2015). Hence, Jiang and Zhuang (2011) apply different extreme value distributions to BA of Canadian forests, and differentiate wildfires according to the type of forest (Boreal vs Taiga) and the source of ignition (human-caused vs lightning). Hernandez et al. (2015) apply the GEV distribution to BA and fire radiative powers obtained using remote sensing techniques over a large box covering the Mediterranean Basin.

This work aims at computing the return periods of wildfire BA in southern France using EVT. Specifically, we question the efficiency of the new fire policy set up in 1994 to reduce the likelihood of very large wildfires. Indeed, even if this policy is probably efficient for reducing the mean BA, its real effect on the largest fires is still unknown. To this aim, our EVT framework is implemented in an explicit non-stationary context adapted to our case study (Fréjaville and Curt, 2015), in time, for two distinct periods (1973 to 1994, and 1995 to 2016), and also in space, to account for three distinguished pyroclimatic sub-regions. In addition, a Bayesian approach (e.g. Robert, 1994; Gelman et al., 2013) is used to assess uncertainties in return periods and related return levels. This allows determining if the highlighted changes are actually significant. We use BA from the Prométhée fire database as a surrogate for fire risk, based on the assumption that the largest fires generate the highest



impacts and costs for suppression and are among the most devastative (Lahaye et al., 2018). Our final goal is to provide risk managers with crucial information on the future likelihood of very large fires in southern France.

## 2 Materials and methods

### 2.1 Study area and pyroclimatic regions

This study covers an area of 80,500 km$^2$ located in the South-East of France (Fig. 1). Fig. 1A presents the main geographical regions which can be classified as Mediterranean lowlands (Provence, Languedoc-Roussillon, Maritime Alps), hinterlands and foothills (Southern Alps, Cévennes), and mountainous areas (the Alps, Corsica, Massif Central and eastern Pyrénées) which corresponds to high elevations (> 1000 meters a.s.l.). Mediterranean areas have a fairly dry and warm climate (mean rainfall < 700 mm.yr$^{-1}$ and temperature > 13°C) and, as a consequence, they are conducive to fire activity. The frequency of wildfires

is smaller in hinterland and mountain areas due to a high mean rainfall (>700 mm.yr$^{-1}$) and medium temperatures (<13°C). Strong and dry winds also increase the summer fire activity along the Mediterranean coastline (fuel dryness, larger fire rate of spread Fréjaville and Curt, 2015).

This study area encompasses different bioclimates, different vegetation fuels, and different levels of human activity (which generate about 95% of ignitions), leading to different fire activities. For this reason we define three main homogeneous sub-

regions based on fire activity and bioclimatic variables, so-called pyroclimatic regions (PCr, adapted from Fréjaville and Curt, 2015, see Fig. 1C). The PCr-1 contains most of fire hotspots of maximal fire activity with many fires (notably the largest ones) and large amounts of BA. It includes Corsica, Provence and the Maritime Alps and concerns 64% of all fires and 67% of the total BA in southern France. In addition, it contains 63% of the largest fires (> 100 ha.) and 74% of the total BA of these fires. This is due to the combination of high fire weather danger, medium to high fuel amount and connectivity in the landscape (Curt

et al., 2013), and to very high density of human-caused ignitions by negligence, by accident or intentional (Curt et al., 2016). This region is undoubtedly prone to large fires, which occur mostly in summer. The PCr-2 has medium to high fire activity and covers the Mediterranean hinterland and mid-elevation mountains. The PCr-3 has low fire activity and few large fires. It corresponds to high mountains with a wet and cool climate.

### 2.2 Fire database and BA thresholds

We use the national georeferenced fire database called 'Prométhée' (Prométhée, 2016) which contains information on more than 106,000 wildfires. It corresponds to wildfire characteristics recorded by fire crews and foresters after each intervention from 1973 to 2016. Prométhée gives standardized information on each fire, including the day and hour of ignition, its location on a $2 \times 2$ km grid (with a total of 20,142 pixels), and the final BA in ha.

Figure 2 presents the temporal evolution of the number of large fires (>100 ha.) for each pyroclimatic region. Clearly, the

fire activity is the highest in PCr-1, and PCr-3 has a limited number of large fires in comparison. From these time series, it is also evident that the fire policy set up in 1994 reduced the number of large fires in all regions.



**Figure 1.** Maps of the study area. A. Geographical map. B. Occurrence of large fires (BA > 1000 ha.). C. Density of the number of fires on a $2 \times 2$ km grid and location of the three pyroclimatic regions.

## 2.3 Extreme Value Theory

Extreme Value Theory (Coles, 2001) indicates that, under some mild conditions, the Generalized Extreme Value distribution (or GEV) is the limiting distribution of maxima applied on very large blocks. Grounding on this strong mathematical result, we assume that wildfire samples for each year are sufficiently large, so that annual maxima of BA can be considered to follow a

5  GEV distribution. This has already been shown to be adequate in a very large number of applications, for example in hydrology (Katz et al., 2002; Papalexiou and Koutsoyiannis, 2013), finance and insurance (Embrechts et al., 1997), physics (Fortin and Clusel, 2015), mountain hazards (Gaume et al., 2013; Favier et al., 2016; Nicolet et al., 2016) or coastal engineering (Mazas and Hamm, 2011) to cite a few of them. The cumulative distribution function (cdf) of the GEV is:





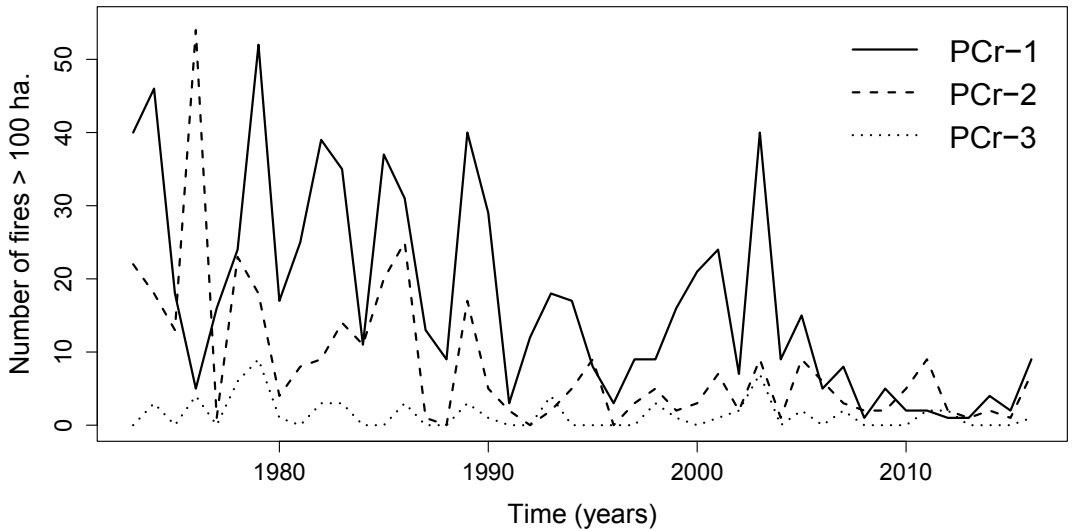

**Figure 2.** Number of large fires (>100 ha.) for each pyroclimatic region, as a function of time.

$$
F(x|\boldsymbol{\theta}) = \begin{cases} \exp\left\{-\left(1+\xi(\frac{x-\mu}{\sigma})\right)^{-1/\xi}\right\} & \text{if } \xi \neq 0, \\ \exp\left\{-\exp\left(-(x-\mu)/\sigma\right)\right\} & \text{if } \xi = 0, \end{cases} \tag{1}
$$

where $\boldsymbol{\theta} = \{\mu, \sigma, \xi\}$ is the set of parameters of the GEV distribution, $\mu$ being the location parameter, $\sigma$ the scale parameter and $\xi$ the shape parameter. When $\xi < 0$, the distribution has an upper tail (the bound cannot be exceeded) which is equal to $\mu - \sigma/\xi$. When $\xi$ becomes large, the distribution tail gets heavier and very large value of $x$ (here the BA) are more likely. The

5  corresponding density function is:

$$
f(x|\boldsymbol{\theta}) = \begin{cases} \frac{1}{\sigma}\left(1+\xi(\frac{x-\mu}{\sigma})\right)^{-1/\xi-1}\exp\left\{-\left(1+\xi(\frac{x-\mu}{\sigma})\right)^{-1/\xi}\right\} & \text{if } \xi \neq 0, \\ \frac{1}{\sigma}\exp\left(-\frac{x-\mu}{\sigma}\right) \times \exp\left\{-\exp\left(-\frac{x-\mu}{\sigma}\right)\right\} & \text{if } \xi = 0. \end{cases} \tag{2}
$$

For the GEV distribution, the quantile $Q(p|\boldsymbol{\theta})$ corresponding to a probability $p$ and a set of parameters $\boldsymbol{\theta}$ is easily obtained:

$$
Q(p|\boldsymbol{\theta}) = \begin{cases} \mu + \sigma\{(\log(1/p)^{-\xi}) - 1\}/\xi & \text{if } \xi \neq 0, \\ \mu + \sigma \log(1/\log(1/p)) & \text{if } \xi = 0. \end{cases} \tag{3}
$$

For a return period $T$ (e.g. $T$=20 years), the return period is thus expressed by $Q(p = 1 - 1/T|\boldsymbol{\theta})$.





## 2.4 Bayesian inference and Monte Carlo simulation

In this study, statistical inference is performed using Bayesian methods (see, e.g., monographs by Robert, 1994; Gelman et al., 2013). A Bayesian analysis combines the information in the data represented by the likelihood function with prior knowledge about the parameter. Parameter estimation is made through the posterior distribution which is computed using Bayes' Theorem

$$\Pi(\boldsymbol{\theta}|\boldsymbol{D}) \propto \Pi(\boldsymbol{D}|\boldsymbol{\theta}) \times \Pi(\boldsymbol{\theta}), \tag{4}$$

where $\Pi(\boldsymbol{\theta}|\boldsymbol{D})$ is the posterior distribution of the GEV parameters $\boldsymbol{\theta}$, $\Pi(\boldsymbol{D}|\boldsymbol{\theta})$ is the likelihood function, and $\Pi(\boldsymbol{\theta})$ is the prior distribution of $\boldsymbol{\theta}$. In this expression, we assume that the normalizing constant of the posterior is not known.

The likelihood function $\Pi(\boldsymbol{D}|\boldsymbol{\theta})$ is the probability corresponding to the logarithm of the maximum BA $\boldsymbol{D}$ given the parameters $\boldsymbol{\theta}$. Here, we assume that the logarithm of these maxima are independently distributed (i.e. the maximum BA in a year $y$ does not influence the maximum of the year $y+1$) and that they follow a GEV distribution. The likelihood is thus given by:

$$\Pi(\boldsymbol{D}|\boldsymbol{\theta}) = \prod_{i=1}^{N} f(x_i|\mu, \sigma, \xi), \tag{5}$$

where $x_i, i, \ldots, N$ are the the logarithm of the maximum BA observed during $N$ years. Here, following (Hernandez et al., 2015), we choose to work with the logarithm of the maximum BA in order to solve scale issues. Indeed, raw BA are two-dimensional data and have very skewed distributions, which often lead to extreme value distributions with an infinite variance (when $\xi > 0.5$, see Moritz, 1997; Alvarado et al., 1998; Jiang and Zhuang, 2011). Applying a log-transformation to maximum BA is a convenient way to avoid this problem.

The prior distribution $\Pi(\boldsymbol{\theta})$ represents our *a priori* knowledge about the parameters $\boldsymbol{\theta}$. Here, we choose:

- Location parameter $\mu \sim N(0, 100000)$: vague prior,

- Scale parameter $\log(\sigma) \sim N(0, 100000)$: vague prior,

- Shape parameter $(\xi + 0.5) \sim Beta(6, 9)$: With this prior, $\xi$ lies within $[-0.5, 0.5]$ and values greater than 0.3 are considered unlikely. Martins and Stedinger (2000) motivate this choice for several practical reasons. First, the variance of the GEV distribution is infinite when $\xi > 0.5$ and the skewness coefficient is infinite when $\xi > 1/3$. Second, numerous geophysical applications show that, generally, $\xi$ lies within this interval.

- The joint prior distribution is $\Pi(\theta) = \Pi(\mu) \times \Pi(\sigma) \times \Pi(\xi)$, i.e. the prior distributions are considered independent.

The posterior distribution $\Pi(\boldsymbol{\theta}|\boldsymbol{D})$ can be sampled using a Markov Chain Monte Carlo (MCMC) algorithm (Gilks et al., 1995; Robert and Casella, 2004). Here, we use the Metropolis-Hastings algorithm (Metropolis et al., 1953). The multivariate normal proposal distribution is scaled using the covariance matrix of the parameter estimates when the maximum-likelihood method is applied. For each estimation, we produce a burn-in sample of size 100,000, and the retained sample used to represent the posterior distribution is of size $M = 10,000$, for which a thinning interval of 10 is applied in order to reduce the autocorrelation inherent in MCMC chains produced with the Metropolis-Hastings algorithm.





Draws from the posterior distribution $\Pi(\boldsymbol{\theta}|\boldsymbol{D})$ can be used to estimate other unknown quantities. Specifically, the predictive distribution of a quantity $y = g(\cdot|\boldsymbol{\theta})$, function of some arguments $\cdot$ and depending on the GEV parameters $\boldsymbol{\theta}$, is expressed as (see Gelman et al., 2013, Eq. 1.4):

$$\Pi(y|\boldsymbol{D}) = \int_{\theta} g(\cdot|\boldsymbol{\theta})\Pi(\boldsymbol{\theta}|\boldsymbol{D})d\boldsymbol{\theta}. \tag{6}$$

Hence, it fairly propagates the uncertainty related to parameter estimation on the quantities of interest. However, a closed expression of the predictive distribution can rarely be obtained, and it is often estimated using the draws from the posterior distribution:

$$\hat{\Pi}(y|\boldsymbol{D}) = \frac{1}{M}\sum_{i} g(\cdot|\boldsymbol{\theta}^{(i)}), \tag{7}$$

where $\boldsymbol{\theta}^{(i)}$ is the $i^{th}$ draw from the posterior distribution. For example, the predictive distribution of a quantile $q = Q(p|\boldsymbol{\theta})$

(see Eq. 3) can be estimated by applying Eq. 7 with $g(\cdot|\boldsymbol{\theta}^{(i)}) = Q(p|\boldsymbol{\theta}^{(i)})$, i.e.:

$$\hat{\pi}(q) = \frac{1}{M}\sum_{i} Q(p|\boldsymbol{\theta}^{(i)}). \tag{8}$$

Likewise, the predictive density of a maximum log-BA $\tilde{x}$ can be estimated with the expression $\hat{\pi}(\tilde{x}) = \frac{1}{M}\sum_{i} f(\tilde{x}|\boldsymbol{\theta}^{(i)})$, which can be used in turn to estimate specific probabilities, such as the exceedance probability of some critical log-BA $x_c$. For example, the probability of having a BA $x$ exceeding $10,000$ ha. in a year is obtained as:

$$\Pr(x > 10,000) = \int_{log(10,000)}^{\infty} \hat{\pi}(\tilde{x})d\tilde{x}, \tag{9}$$

where $\tilde{x}$ is integrated over all values exceeding $10,000$ ha., on the logarithm scale.

## 2.5   Measures of similarity between two distributions

Numerous statistical tests aims at testing an assumption of equality between two distributions (for example the test of Kolmogorov-Smirnov). These tests are adequate when two samples are compared and when we want to test if the sampling distributions

can be considered as equivalent. However, it must be noticed that these statistical tests are very powerful with large samples, which means that in this case, a small difference will be considered as highly significant. As a result, comparing posterior distributions with these statistical tests generally result in a rejection of the assumption of equality, even when the distributions are very similar.





As an alternative, different measures have been proposed by statisticians in order to quantify the similarity between two distributions, or, in other words, how much two distributions overlap (e.g. Malalanobis distance, Kullback-Leibler distance). Bhattacharyya (1946) proposes a coefficient whose expression is given by:

$$BC(p,q) = \int \sqrt{p(x)q(x)}\,dx, \tag{10}$$

where $p(x)$ and $q(x)$ are the probability density functions to be compared. We have $0 \leq BC \leq 1$, with $BC = 0$ indicating that the distributions do not overlap and $BC = 1$ indicating a complete similarity.

For the sake of comparison with statistical tests, when this coefficient is applied to two normal distributions $N(0,1)$ and $N(2,1)$ (shift of two standard deviations), it has a value of 0.61. Thus, it can be considered than two distributions are similar when the Bhattacharyya coefficient exceeds 0.61. This reference value, while subjective, gives a more precise idea of the interpretation of the Bhattacharyya coefficient.

## 3   Results

### 3.1   Relevance of the GEV distribution

For each of the thee regions PCr-1, PCr-2 and PCr-3, we collect the annual maxima of BA. Following the methodology described in Section 2, we fit a GEV distribution to the logarithm of annual maxima of BA for the periods 1973–1994 and 1995–2016, in order to assess the impact of the fire policy established in 1994. Taking the mean of the posterior distributions as point estimates, Figure 3 shows the QQ-plots of the fitted GEV distributions, for each region and each period. This figure compares observed maxima and quantiles from the fitted GEV distribution corresponding to empirical probabilities of observed maxima. Points are close to the 1:1 line which indicate that these adjustments are reasonable.

### 3.2   Change in fire extremes before/after 1994

Posterior distributions of parameters $\mu$, $\sigma$ and $\xi$, before and after 1994 and for each region, are shown in Figure 4. Large differences are obtained for some parameters and some regions. For example, for regions PCr-1, the location parameter $\mu$ has clearly decreased between the two periods, indicating that the fitted distribution has globally shifted toward smaller BA after 1994. The scale parameter has decreased in regions PCr-2 and PCr-3, and slightly increased in region PCr-1. The shape parameter exhibits a slight increase in all regions. However, parameter uncertainty is large and the posterior distributions before/after 1994 overlap, which limits the interpretation of this increase.

Table 1 quantifies these differences using Bhattacharyya coefficient. As indicated above, this coefficient measures the similarity between two distributions. It is equal to 0 when the distributions do not overlap and 1 if they are identical. Table 1 indicates that, for the region PCr-1, posterior distributions for $\mu$ before and after 1994 are clearly different whereas they are roughly similar for the parameters $\sigma$ and $\xi$, confirming the global shift of the distribution mentioned above. For the region PCr-2, only $\sigma$ has a significant change before/after 1994, while no difference is revealed for the region PCr-3.





**Figure 3.** QQ-plot of the fitted GEV distributions for each region (PCr-1, PCr-2 and PCr-3) and each period (1973–1994 and 1995–2016).
Point estimates of the GEV parameters are obtained as the mean of the posterior distributions. The BA (ha.) is indicated on a log-scale.

For a return period $T$ (e.g. $T = 100$ years), the corresponding BA is directly obtained, for a set of parameters $\boldsymbol{\theta}$, using the
expression (3). With the Bayesian approach, we can obtain the predictive distribution of these return levels by applying Eq. (3)
to all MCMC samples of parameters $\boldsymbol{\theta}^{(i)}, i = 1, \ldots, M$ drawn from the posterior distribution (see Eq. 8).

Figure 5 compares the predictive distributions of BA corresponding to return periods ranging from 2 years to 50 years, for
5   periods before (in red) and after 1994 (in blue), for the three regions. For all regions, and particularly for region PCr-1, BA
corresponding to the smallest return periods (e.g. 2 years) have decreased, which demonstrates the efficiency of the fire policy



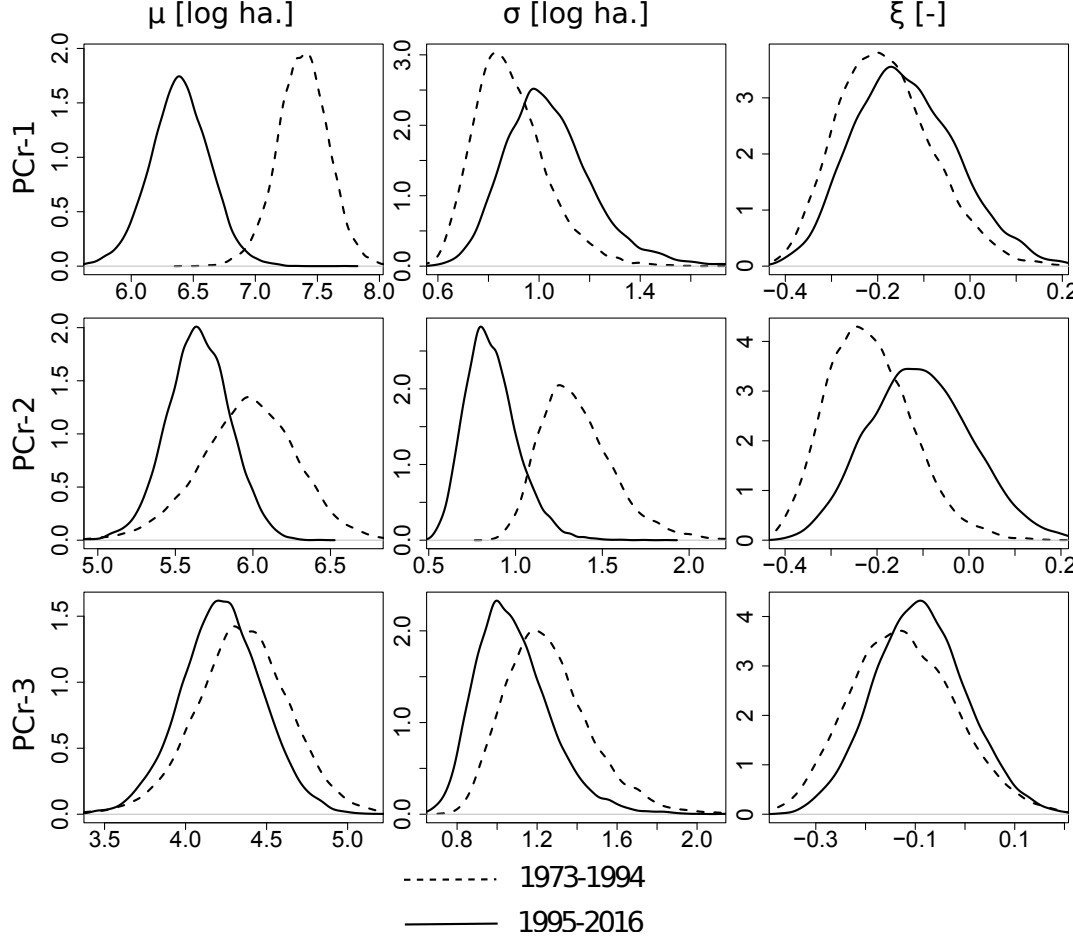

**Figure 4.** Comparison of posterior distribution of the GEV parameters, before (dashed line) and after 1994 (solid line): location parameter $\mu$ (log ha.), scale $\sigma$ parameter (log ha.) and shape parameter $\xi$ (dimensionless), for each region.

**Table 1.** Bhattacharyya coefficients applied on posterior densities of the GEV parameters, before/after 1994, for each region. Bold values indicate coefficients below the reference value of 0.61.

| Zone | $\mu$ | $\sigma$ | $\xi$ |
|------|------|------|------|
| PCr-1 | **0.11** | 0.89 | 0.98 |
| PCr-2 | 0.79 | **0.36** | 0.87 |
| PCr-3 | 0.97 | 0.91 | 0.98 |

established in 1994. However, for region PCr-1, the 90% credible intervals overlap for higher return periods (e.g. 50 years). For this critical region, considering the large uncertainties associated to high return periods, no clear conclusion can be drawn





in terms of decrease/increase of extreme return levels. The corresponding Bhattacharyya coefficients are given in Table 2, and indeed show that return levels have only changed significantly for return periods smaller than 10 years. For region PCr-2, the decrease of the return levels is more important, even for higher return periods (see Table 2). Finally, there is no noticeable change for region PCr-3.

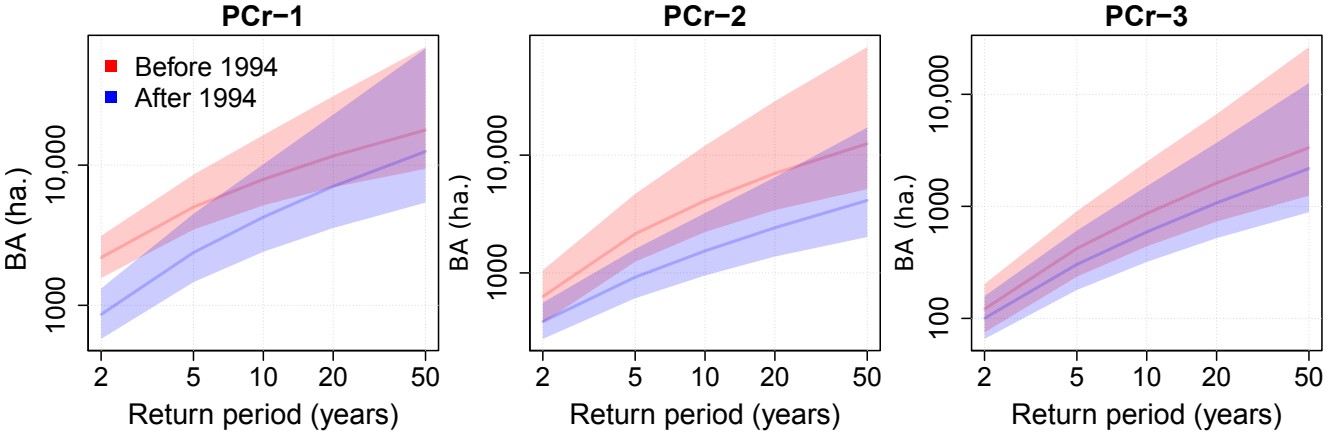

**Figure 5.** Comparison of return levels, before/after 1994, for each region. 90% credible intervals are shown and median return levels are indicated in plain lines. BA (ha.) is indicated on a log-scale.

**Table 2.** Bhattacharyya coefficients applied on different return levels, before/after 1994, for each region. Bold values indicate coefficients below the reference value of 0.61.

| Zone | 2 years | 5 years | 10 years | 20 years | 50 years |
|------|---------|---------|----------|----------|----------|
| PCr-1 | **0.16** | **0.46** | 0.66 | 0.80 | 0.90 |
| PCr-2 | 0.65 | **0.45** | **0.49** | **0.57** | 0.68 |
| PCr-3 | 0.95 | 0.92 | 0.92 | 0.94 | 0.96 |

5    The probability of having very large fires (>10,000 ha.) in a year are estimated using Eq. 9 and are given in Table 3. These estimates indicate a clear decrease of this probability before/after 1994 in region PCr-1 (from 0.076 to 0.037) and PCr-2 (from 0.040 to 0.008). In PCr-2, the probability of having such large fires remains small (from 0.009 to 0.006).

**Table 3.** Estimated probability of having very large fires (>10,000 ha.) in a year, by area and for periods 1973–1994 and 1995–2016.

|  | 1973–1994 | 1995–2016 |
|------|-----------|-----------|
| PCr-1 | 0.076 | 0.037 |
| PCr-2 | 0.040 | 0.008 |
| PCr-3 | 0.009 | 0.006 |



### 3.3 Return levels for the recent period (1995–2016)

Table 4 reports the BA corresponding to high return periods (20 and 50 years), for the recent period 1995–2016. Median return levels correspond to BA smaller than 5,000 ha. in regions PCr-2 and PCr-3 but not in region PCr-1 for which BA corresponding to 20 and 50 years return periods are very large (respectively 7,050 ha. and 12,510 ha.). However, very high uncertainties are

associated to these return levels. For example, the 90% credible interval corresponding to the 50-year return level of 12,510 ha. in PCr-1 is very wide (from 5,400 ha. to 68,000 ha.), which limits the interpretation of this estimate.

**Table 4.** BA (ha.) corresponding to different return periods (years), by area and for the recent period (1995–2016). Intervals between brackets indicate 90% credible intervals of the corresponding median return levels.

|  | 20 years | | 50 years | |
| --- | --- | --- | --- | --- |
| PCr-1 | 7,050 | [3,560; 22,910] | 12,510 | [5,400; 68,000] |
| PCr-2 | 2,420 | [1,380; 6,490] | 4,130 | [2,020; 17,160] |
| PCr-3 | 1,070 | [520; 3,700] | 2,170 | [890; 12,620] |

### 4 Discussion: using return levels to manage fire risk

This study provides for the first time a model of fire return periods in southern France, taking into account the non-stationary of fire activity in space and time. Extreme value theory provides arguments in favor of the application of extreme value

distributions (Coles, 2001). We show that the GEV distribution adequately fits the annual maxima of BA for each region and each period. In addition, we properly evaluate the uncertainty related to high return level using a Bayesian approach to estimate the parameters of the GEV distribution. As a result, the significance of potential changes can be assessed.

It appears that very large fires (>10,000 ha.) are currently rare but not improbable in region PCr-1 (probability of 0.037), and less likely in regions PCr-2 (probability of 0.008) and PCr-3 (probability of 0.006), see Table 3. Hence, a major finding of

this study is that, despite the new suppression-oriented policy set up in 1994, very large fires can still occur in southern France, and especially in the regional hotspots of Corsica and Provence. Indeed, in region PCr-1, the distribution of maximum BA has globally shifted toward smaller values between the periods 1973–1994 and 1995–2016 (see Fig. 5), but the current distribution is also more skewed. As a consequence, the BA corresponding to a return period of 50 years has not significantly decreased.

These findings provide information for leveraging fire risk. In the most fire-prone areas of France (Corsica and Provence),

it is less and less doubtful that several ongoing changes will create opportunities for extreme fire events, as in many European countries (Tedim et al., 2018). Indeed, a combination of climate change, fuel accumulation and increasing human pressure on ignitions in urbanized areas is the root cause for a new generation of wildfires characterized by extreme behavior (Costa et al., 2011). These new types of devastative fires are increasingly observed in southern Europe, such as in Portugal in 2017 (the Pedrógão Grande megafire burned 45.000 ha in June, 2017, and resulted in 66 fatalities) or in France in 2016 and 2017.

In example, the Rognac-Vitrolles fire burned 2700 ha. (August, 2016) and destroyed 39 habitations and a school. 560 trucks



and 1800 firemen were necessary to suppress the fire. Many of those fires are beyond the present suppression capacity (Lahaye et al., 2018). They often display eruptive or erratic behaviors, and they are able to propagate through landscape mixes in the wildland-urban interfaces and in cultivated areas. Recent studies in southern France have shown that new combinations of prolonged droughts, heatwaves and windy periods lead to a higher frequency of such extreme fire events (Ruffault J. et al.,

2016; Ruffault et al., 2016, 2018). In areas prone to extreme wildfire events, maintaining the pre-positioning of firemen in the field and massive aerial and ground forces for suppression is necessary every year. However, this may not be sufficient to control all fires during certain periods of very high hazard. In this context, managing landscapes and fuels, limiting unwanted ignitions and the participation of the public to self-protection (i.e. tackling the root causes of fire risk) are key to leverage fire risk on the long-term. Concerning other pyroclimatic regions where fire activity is lower, the probability of very large fires

is more limited. In mid-elevation hinterland (PCr-2) and in mountains (PCr-3), only a dozen of fires larger than 100 hectares were recorded each year in average since the 1990s. The ongoing climate changes favor an extension of areas with high fire weather danger towards the North and in mountains (Dupire et al., 2017) and the season favorable to fire activity becomes longer. However, ignitions remain limited and fuels are rarely fully dry. This limits the probability of having very large fires. In these regions, the aerial surveillance by air tackers has been increased in order to detect fires as soon as possible.

Finally, it must be remembered that, in this study, we use the occurrence and BA of very large fires as a surrogate for extreme fire danger because no reliable and exhaustive database provides information on the real impacts of fire. This deserves discussion. On average, the largest fires actually have huge impacts on humans (including fire crews) and the environment because they are likely to have an effect on a large number of assets (Fernandes et al., 2016). This is especially true along the Mediterranean coastline which is densely populated, densely urbanized, and very attractive for tourists. In addition, the largest

fires need high expenditure to be suppressed (Lahaye et al., 2018). In this context, BA is likely the most appropriate surrogate for extreme fire events. However, a perspective is to better estimate their impacts by comparing the pre- and postfire biomass in ecosystems, and by building databases dedicated to impacts on people and infrastructures.

## 5 Conclusion

Extreme fire events (i.e. very large fires generating high human, economic and ecological damages) are a growing issue in

southern Europe and almost worldwide. They have a disproportionate impact on the medias and they challenge the suppression-oriented policies because they question our ability to control or prevent them in the long term. In France, firefighting accounts for two-thirds of the total budget but it cannot suppress all the large fires as demonstrated notably in 2003, 2016 and 2017 (Chatry et al., 2010). Many of them are erratic, fast growing, or convective (Lahaye et al., 2018) and out of control by firemen. They may belong to a new generation of fires promoted by global changes (Costa et al., 2011), which cause most of the

accidents or fatalities for fire crews. This study demonstrates that even if the fire policy established in 1994 in southern France is undoubtedly successful, changes for BA corresponding to large return periods appear as barely significant. 50-year BA remain important, especially in region PCr-1 (>10,000 ha.), which includes Corsica, Provence and the Maritime Alps. Based





on these results, we propose solutions for leveraging them in a sustainable way in the future. Also, the rigorous proposed methodology which combines extreme value analysis and Bayesian tools may be useful for other case studies worldwide.

## 6 Acknowledgments

Irstea - UR ETGR is member of Labex OSUG@2020. Irstea - RECOVER is member of Labex OT-MED.





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
