# Peer review of "Has fire policy decreased the return period of the largest wildfire events in France? A Bayesian assessment based on extreme value theory"

_Natural Hazards and Earth System Sciences, 2018_

## Referee Comment (RC1) · Anonymous Referee #1 · 27 Jul 2018

In the paper under evaluation, the authors compute return levels of wildfire area burned in southern France in two different periods and for three pyroclimatic regions (PCr-1, PCr-2 and PCr-3). The paper comprises five sections. After the introduction, the authors introduce in Section 2 the fire database and the statistical methods. Results and discussion are presented in Section 3 and 4, respectively. Finally, Section 5 is devoted to conclusions.

The overall impression of the paper is that, although the topic of research is interesting the paper is not publishable in the present form. Below are some comments:

Throughout the text "burned area" should read "area burned"

[Figure]

The authors should better explain what is the benefit of using a Bayesian framework within this context as compared with other competitor approaches. Such explanation will strengthen the paper.

p.2, l.30-31: the authors stress that "...our EVT framework is implemented in an explicit non-stationary context adapted to our case study". The authors should better explain, in a convincing way, to which "explicit non-stationary context" are they referring to. This is not clear neither from the introduction nor from section 3.

p.2, l.33-34: replace "...to assess uncertainties in return periods and related return levels." by "...to assess uncertainties in return levels."

p.2, l. 34: the sentence "This allows determining if the highlighted changes are actually significant." is not understandable as it is. Please elaborate.

p.4, l.3: replace "(or GEV)" by "(or GEV, in-short)".

p.4, l.3-5: the sentence "Grounding on this strong mathematical result, we assume that wildfire samples for each year are sufficiently large, so that annual maxima of BA can be considered to follow a GEV distribution." is not understandable as it is. Please elaborate.

p.4, l.5-9: the sentence "This has already been shown to be adequate...to cite a few of them" adds anything to the paper and should be removed from the text.

p.5, l.1: the authors should include in eq (1) the range of variation of x and theta.

p.5, l.3: the authors should also provide an explanation to the case xi=0.

p.5, l.3: remove "(the bound cannot be exceeded)" from the text.

p.6, l.7: the sentence "In this expression, we assume that the normalizing constant of the posterior is not known." should read "It is assumed that the normalizing constant of the posterior distribution is unknown.".

p.6, l.8-9: the sentence "The likelihood function...parameters theta" is not understandable as it is. Please elaborate.

p.6, l.13-14: the sentence "Indeed, raw BA are twodimensional data and have very skewed distributions, which often lead to extreme value distributions with an infinite variance" is not understandable as it is. Please elaborate.

p.6, l.24: the authors should better explain the reasoning for considering the joint prior distribution as being the product of the marginals; in other words, what is the justification for assuming that the GEV parameters are independent? Furthermore, in the joint prior expression, the term \Phi(\sigma) should read \Phi(\log\sigma).

p.7, l.18: remove "(for example the test of Kolmogorov- Smirnov)"

p.8, l.3: the authors should better justify the choice of the Bhattacharyya coefficient for measuring the distance between distribution functions. Furthermore, the authors should also highlight the advantages and limitations of such coefficient when compared with other competitors such that Malalanobis distance or the weighted L2-Wasserstein distance, among others.

Section 3: the authors should carry out the same analysis but considering several distance measures and to compare the results. This will strengthen the paper.

p. 6-7: the authors should clearly explain the steps for implementing their algorithm. In my opinion the authors fail in explaining their procedure in a way understandable to the reader. Furthermore, details on how their method is implemented in practice are largely omitted which difficult the procedure understanding. For example, nothing is said about the convergence of the MCMC method. Did the authors checked the convergence of the algorithm? Did the authors checked for the presence of metastability? The authors also should include a table displaying the acceptance rates for the GEV parameters within the MCMC algorithm.

p.8, l.20: the sentence "Posterior distributions of parameters..." should read "Posterior

distributions estimates for the extreme value parameters..."

p.8, l.20: replace "sigma" by "log(sigma)"

p.9, l.1: replace "...corresponding BA is ...." by "...corresponding BA estimate is ...."

p.9, l.2: replace "...predictive distribution of these..." by "...predictive distribution estimate of these..."
* * *

---

## Referee Comment (RC2) · Anonymous Referee #2 · 27 Jul 2018

Major comments: The manuscript of Evin et al. attempts to demonstrate statistically that current fire policies in southern France had an effect on large fires burnt area with a return interval of 5 years but not on that of 50 years. They conclude that massive investments in aerial and ground forces are not sufficient to control large fires during extreme fire season (like the 2017 one) and that other strategies should be integrated (e.g. landscape management, self-protection) to leverage fire risk on the long-term. I appreciate the effort to demonstrate analytically a common believe (i.e. fire suppression policies are not sufficient) usually addressed with a qualitative approach or simple descriptive statistics. Although I agree with the general thesis that current fire policies are not yet able to manage large fire seasons like summer 2017 or the ongoing

2018 (and I fully support alternative strategies proposed in the manuscript), I'm not convinced that this experiment provides sufficient evidence that fire policies introduced in France since 1994 are inadequate to manage large fire seasons. Indeed, despite the statistical tests used in the study are quite sophisticated and applied correctly, I have a doubt about modelling fire return periods of 20 to 50 years with a time series of 21 years. Indeed, model uncertainty is very high (Table 4) and at the end of the results section authors state that this uncertainty limits the interpretation of the estimate. Consequently, the key point of the discussion and conclusion (that current fire policy implemented in France is not effective against large fires with a return interval of 50 years) is based on this single uncertain result. The statement at the beginning of the discussion, referring to the sole fire prone region PCr-1, i.e. "...the BA corresponding to a return period of 50 years has not significantly decreased" does not take into account limitations in the analyses. Consequently, the discussion that follows appears to force results interpretation toward a thesis (although, I repeat, it is a thesis that I fully support). In addition, it is not clear to me if differences in other fire regime drivers such as climate, fuel flammability and landscape connectivity where considered in the model when comparing 1973-1994 and 1995-2016 periods. Indeed, if you want to test the "fire policy" driver the the model should account for the variability expalined by other relevant drivers. Note that in Figure 2, after 1994 the sole fire peak in the number of fires>100 ha reaching a level similar to ones before 1994 corresponds to the 2003 fire season, i.e. the major climate anomaly hitting southern France during the period of analysis. Notably, one of the author in a previous similar paper (Curt and Frejaville 2017; DOI: 10.1111/risa.12855) stresses the increase in fire weather index, human pressure and fuel coverage in the second studied period.

Minor comments: Pg1, LN17, LN18 and throughout the text - Eliminate dots after "ha"

Pg1, LN16-LN20 – report initial and end period for fire statistics listed in this paragraph

Pg2, LN1 - Include here other relevant references, e.g. Moreira et al. 2011 (DOI: 10.1016/j.jenvman.2011.06.028), Fernandes et al. 2013 (DOI: 10.1890/120298)

Pg2, LN3-LN5 – I believe here is missing a major driver of the burnt area in southern Europe, i.e. cultural and socio-economic aspects affecting landscape management (i.e., type of urbanization, agriculture and forestry, land control, use of fire, type of post-fire management) which in turn contribute to determine fire likelihood and burnt area. Note that this is supported also by authors at Pg. 3, LN15

Pg2, LN9-LN11 – While I agree knowing the return period of large fires it is useful to governmental agencies and reinsurance companies to evaluate the cost of future fires, I do not believe it is useful to the dimensioning of fire crews during an extreme fire event (this is something decided in real time once the ignition point, the fire weather, potential fire trajectories and values at risk are known). Rather, as the return period of a flood is useful to the dimensioning of infrastructures such as embankments of a river (a similarity used by authors at LN 6-7), the return period of a large fire in a valley is useful to the dimensioning of fuel management measures, e.g. how many fuelbreaks, where they must be located in the landscape, how much large they must be, which is the interval between fuel treatments to maintain fuelbreaks before large fires return, which in turn determine management costs and consequently the number of fuelbreaks I can maintain in a given period.

Pg2, LN14 – later in reading the manuscript I assumed "return levels" the same as "return period", but then I realized it was not the case. However, it is not clear which is the difference between the two. Please clarify here or in the method section

Pg2, LN15 - after "...dedicated studies are available" – Although later in the paragraph authors report several references in relation to methods used to calculate the fire return period, I suggest to insert here 2-3 references to previous studies calculating the large fire return period that author think are very relevant for fire management purposes

Pg2, LN20 – after "Extreme Value Theory" add "(EVT)"

Pg2, LN29 – the fire policy change in 1994 in France appears here for the first time, but it is not clear in what the policy consists, and no references are provided. I would expect

here, or later in the methods, a clear referring to the policy, and some quantitative data (i.e. indicators of changes in comparison to the previous policy, e.g. number of helicopters used during the fire season, annual area treated with prescribed burning) characterizing the policy. A table could be useful to synthetize information

Par 2.3 and 2.4 – Clearly state what $\mu, \sigma, \xi$ indicators means in terms of fire management Pg8, LN24 – I do not see where the "parameter uncertainty" is reported. Include model uncertainty in figure 4?

Pg12, LN2 – what is meant with "median return levels"? If 20 years, change "Table 4 reports the BA corresponding to high return periods (20 and 50 years)" in "Table 4 reports the BA corresponding to median and high return periods (20 and 50 years, respectively)"

Figure 1 – Large fires are defined as > 1000 ha, while in the text is > 100 ha. As regards the figure caption – after "pyroclimatyc regions" include "(numbered circles)", or something in the legend clarifying what colored circles represent

Figure 3 – as the aim of the paper does not focus on statistical and methodological aspects I would move figure 3 to the supplementary material

Table 2 – it is not clear how it is possible to model fire return intervals > 10 years with time series of 21 years (1973-1994 and 1995-2016)

---

## Author Comment (AC1) · 28 Aug 2018

**Authors reply on comments of referee #1**

*The overall impression of the paper is that, although the topic of research is interesting the paper is not publishable in the present form.*

We thank the review for these comments. The paper will be revised to take them into account.

**Comment R1 #1.1.** Throughout the text "burned area" should read "area burned".

Depending upon the sentence, « burned area" will be changed into "area burned, or kept, since both are possible.

**Comment R1 #1.2.** The authors should better explain what is the benefit of using a Bayesian framework within this context as compared with other competitor approaches. Such explanation will strengthen the paper.

A Bayesian framework has several advantages compared to standard frequentist approaches. First, explicit *a priori* assumptions can be made about the model parameters, as has been done for the shape parameter $\xi$ (see p.6, l.20-25 of the current manuscript). Second, Bayesian methods provide a direct assessment of the uncertainty related to the parameter estimation (see, e.g. Figure 4 of the current manuscript). Generally, frequentist methods only provide confidence intervals based on theoretical results which hold true for very large samples (i.e. asymptotically). Third, in Bayesian methods, the uncertainty of a quantity of interest can be quantified by means of the predictive distribution (as explained in p.9 of the current manuscript). This predictive distribution integrates over the posterior uncertainty in the parameters and can be obtained directly from the MCMC samples. Here as well, frequentist methods provide similar theoretical results which hold true under some restrictive conditions.

We agree that these explanations would benefit to the paper, and they will be included in the revised version of the manuscript.

**Comment R1 #1.3.** p.2, l.30-31: the authors stress that "...our EVT framework is implemented in an explicit non-stationary context adapted to our case study". The authors should better explain, in a convincing way, to which "explicit non-stationary context" are they referring to. This is not clear neither from the introduction nor from section 3.

The non-stationary context simply refers to the fact that the frequency analysis is applied for two distinct time periods, and three different regions. Therefore, we can consider that our EVT framework is non-stationary in time and in space. However, we agree that this was not clear in the manuscript and this will be better explained in the revised version.

**Comment R1 #1.4.** p.2, l.33-34: replace "...to assess uncertainties in return periods and related return levels." by "...to assess uncertainties in return levels."

Ok, this will be modified.

**Comment R1 #1.5.** p.2, l. 34: the sentence "This allows determining if the highlighted changes are actually significant." is not understandable as it is. Please elaborate.

This sentence indicates that the comparison of the results between the different time periods and regions can provide an assessment of the differences, for example how

significant are then changes in return levels between 1973-1994 and 1995-2016. This will be better explained in the revised version.

**Comment R1 #1.6.** p.4, l.3: replace "(or GEV)" by "(or GEV, in-short)".

Ok, this will be modified.

**Comment R1 #1.7.** p.4, l.3-5: the sentence "Grounding on this strong mathematical result, we assume that wildfire samples for each year are sufficiently large, so that annual maxima of BA can be considered to follow a GEV distribution." is not understandable as it is. Please elaborate.

Extreme Value Theory indicates that maxima of regular large blocks follow a GEV distribution. This means that if these maxima are taken from very large regular samples, these maxima should follow a GEV distribution. In our case, these samples are the burned areas for each year. The database contains 106,000 wildfire records, and therefore we assume that wildfire samples for each year are sufficiently large. An additional sentence in the revised manuscript should clarify this point.

**Comment R1 #1.8.** p.4, l.5-9: the sentence "This has already been shown to be adequate...to cite a few of them" adds anything to the paper and should be removed from the text.

In our opinion, these citations are useful and should be kept in the revised version. First, other applications of the EVT show that this theory is now widely spread in applied fields and is handled by many practitioners. Second, these references provide several examples of applications of the EVT and can help the reader to better understand how it works (for example how the block maxima are obtained, how the parameters can be estimated, etc.).

**Comment R1 #1.9.** p.5, l.1: the authors should include in eq (1) the range of variation of x and theta.

We thank the reviewer for this comment, this will be done.

**Comment R1 #1.10.** p.5, l.3: the authors should also provide an explanation to the case xi=0.

When xi=0, the GEV distribution has no bounds, this will be indicated in the revised version.

**Comment R1 #1.11.** p.5, l.3: remove "(the bound cannot be exceeded)" from the text.

Ok, this will be modified.

**Comment R1 #1.12.** p.6, l.7: the sentence "In this expression, we assume that the normalizing constant of the posterior is not known." should read "It is assumed that the normalizing constant of the posterior distribution is unknown."

Ok, this will be modified.

**Comment R1 #1.13.** p.6, l.8-9: the sentence "The likelihood function...parameters theta" is not understandable as it is. Please elaborate.

The likelihood function represents the conditional density of the data **D**, here the logarithm of the maximum BA, given the parameters **θ**, and describes the plausibility of observing the data **D** given **θ**. This will be clarified in the revised manuscript.

**Comment R1 #1.14.** p.6, l.13-14: the sentence "Indeed, raw BA are two-dimensional data and have very skewed distributions, which often lead to extreme value distributions with an infinite variance" is not understandable as it is. Please elaborate.

Raw BA are measures of surface, i.e. two-dimensional data. It creates a scale issue, very large BA values (e.g. 1,000 ha) being a lot larger than large BA values (e.g. 100 ha). This scale issue becomes obvious when the distribution of the raw BA is represented, this distribution being extremely skewed. As a consequence of these very skewed distributions, extreme value distributions with an infinite variance ($xi>0.5$) are often obtained. We will clarify this point in the revised manuscript.

**Comment R1 #1.15.** p.6, l.24: the authors should better explain the reasoning for considering the joint prior distribution as being the product of the marginals; in other words, what is the justification for assuming that the GEV parameters are independent? Furthermore, in the joint prior expression, the term \Phi(\sigma) should read \Phi(\log\sigma).

In the absence of *a priori* information about some type of dependence between the GEV parameters, it is usual to use independent priors (see, e.g., p.174 in Coles, 2001). This does not preclude from quantifying inter-parameter correlation within the joint posterior distribution if there is evidence in the data that such correlation actually exists. We will add this point to the revised manuscript.

**Comment R1 #1.16.** p.7, l.18: remove "(for example the test of Kolmogorov-Smirnov)"

We do not understand why this example of a statistical test for the equality of two distributions needs to be removed. This example of a standard statistical test can help a non-statistician reader to understand this paragraph.

**Comment R1 #1.17.** p.8, l.3: the authors should better justify the choice of the Bhattacharyya coefficient for measuring the distance between distribution functions. Furthermore, the authors should also highlight the advantages and limitations of such coefficient when compared with other competitors such that Mahalanobis distance or the weighted L2-Wasserstein distance, among others. Section 3: the authors should carry out the same analysis but considering several distance measures and to compare the results. This will strengthen the paper.

It is true that many options exist in terms of 'generalized distances", or divergence measures between two distributions. Mahalanobis (1930) first acknowledges that tests of significance are sometimes limited and for this reason he introduced the idea of divergence of two populations. In our case, Mahalanobis distance is not well suited since the sample version assumes a common standard deviation (see McLachlan 1999 p. 24). In this study, the distributions to be compared have different standard deviations (see mu for Pcr-2 in Fig. 4). However, many other alternatives could be applied. For example, the Kullback-Leibler divergence is widely spread, and the weighted L2-Wasserstein distance is another alternative. Different measures have been tested and they lead to identical conclusions. As an example, Table 1 presents the Kullback-Leibler divergence applied on posterior densities of the GEV parameters, before/after 1994, for each region, to be compared with Table 1 of the current manuscript.

In this paper, the Bhattacharyya coefficient is applied mainly because it can be easily normalized between 0 and 1, which facilitates its interpretation. However, we agree

that this choice should be further motivated and discussed in the manuscript. This will be done in the revised version.

**Table 1: Kullback-Leibler divergence applied on posterior densities of the GEV parameters, before/after 1994, for each region. Bold values indicate coefficients below the reference value of 2.16 (for two normal distribution N(0,1) and N(2,1)).**

| Zone | μ | σ | ξ |
|------|------|------|------|
| PCr-1 | **9.17** | 0.46 | 0.08 |
| PCr-2 | 1.73 | **3.26** | 0.53 |
| PCr-3 | 0.15 | 0.40 | 0.09 |

**Comment R1 #1.18.** p. 6-7: the authors should clearly explain the steps for implementing their algorithm. In my opinion the authors fail in explaining their procedure in a way understandable to the reader. Furthermore, details on how their method is implemented in practice are largely omitted which difficult the procedure understanding. For example, nothing is said about the convergence of the MCMC method. Did the authors check the convergence of the algorithm? Did the authors checked for the presence of metastability? The authors also should include a table displaying the acceptance rates for the GEV parameters within the MCMC algorithm.

At p.6 l.25-30 of the current manuscript, we indicate that we use the Metropolis-Hastings algorithm in order to sample the posterior distribution. In details, we use the function MCMCmetrop1R from the MCMCpack package in R software (R 2017). We also indicate that the multivariate normal proposal distribution is scaled using the covariance matrix of the parameter estimates when the maximum-likelihood method is applied. This is achieved using the fgev from the evd package in R.

As indicated in the manuscript, for each estimation, we produce a burn-in sample of size 100,000, and the retained sample used to represent the posterior distribution is of size M=10,000, for which a thinning interval of 10 is applied in order to reduce the autocorrelation inherent in MCMC chains produced with the Metropolis-Hastings algorithm. Such a long burn-in period usually avoids convergence issues for these dimensions (i.e. 3-dimensional parameter vectors). Table 2 presents the acceptance rates for the GEV parameters, and lie between 0.32 and 0.42. These acceptance rates are reasonable and correspond to common requirements. Indeed, acceptance rates between 0.23 and 0.5 are usually advised in order to obtain a fast convergence of the algorithm (Robert and Casella 2004).

The convergence was checked visually. As an example, Figure 1 shows the traces of the sampled parameters and density estimates for each parameter, for the region PCr-3 and period 1995-2016. The trace is regular and no jumps can be observed. This assessment did not reveal any convergence issue.

The details about the implementation of the algorithm will be added to the revised version of the manuscript, as well as Table 2 displaying the acceptance rates for the GEV parameters.

**Table 2: acceptance rates from the Metropolis-Hastings algorithm.**

| Zone | 1960-1994 | 1995-2016 |
|---|---|---|
| PCr-1 | 0.42 | 0.34 |
| PCr-2 | 0.35 | 0.32 |
| PCr-3 | 0.33 | 0.40 |

[Figure]

**Figure 1: Trace of the sampled parameters and density estimates for each parameter, for the region PCr-3 and period 1995-2016.**

**Comment R1 #1.19.** p.8, l.20: the sentence "Posterior distributions of parameters..." should read "Posterior C3 distributions estimates for the extreme value parameters..."

Ok, this will be modified.

**Comment R1 #1.20.** p.8, l.20: replace "sigma" by "log(sigma)"

Ok, this will be modified.

**Comment R1 #1.21.** p.9, l.1: replace "...corresponding BA is ...." by "...corresponding BA estimate is ...."

Ok, this will be modified.

**Comment R1 #1.22.** p.9, l.2: replace "...predictive distribution of these..." by "...predictive distribution estimate of these..."

Ok, this will be modified.

**References**

McLachlan, G. J. 1999. "Mahalanobis Distance." *Resonance* 4 (6): 20–26. https://doi.org/10.1007/BF02834632.

R. 2017. *R: A Language and Environment for Statistical Computing* (version 3.4.0). R Foundation for Statistical Computing. Vienna, Austria: ISBN 3-900051-07-0.

Robert, Christian, and George Casella. 2004. *Monte Carlo Statistical Methods*. Springer Texts in Statistics. New-York: Springer-Verlag.

---

## Author Comment (AC2) · 28 Aug 2018

**Authors reply on comments of referee #2**

**Major comments:**

The manuscript of Evin et al. attempts to demonstrate statistically that current fire policies in southern France had an effect on large fires burnt area with a return interval of 5 years but not on that of 50 years. They conclude that massive investments in aerial and ground forces are not sufficient to control large fires during extreme fire season (like the 2017 one) and that other strategies should be integrated (e.g. landscape management, self-protection) to leverage fire risk on the long-term. I appreciate the effort to demonstrate analytically a common believe (i.e. fire suppression policies are not sufficient) usually addressed with a qualitative approach or simple descriptive statistics. Although I agree with the general thesis that current fire policies are not yet able to manage large fire seasons like summer 2017 or the ongoing 2018 (and I fully support alternative strategies proposed in the manuscript), I'm not convinced that this experiment provides sufficient evidence that fire policies introduced in France since 1994 are inadequate to manage large fire seasons. Indeed, despite the statistical tests used in the study are quite sophisticated and applied correctly, I have a doubt about modelling fire return periods of 20 to 50 years with a time series of 21 years. Indeed, model uncertainty is very high (Table 4) and at the end of the results section authors state that this uncertainty limits the interpretation of the estimate. Consequently, the key point of the discussion and conclusion (that current fire policy implemented in France is not effective against large fires with a return interval of 50 years) is based on this single uncertain result. The statement at the beginning of the discussion, referring to the sole fire prone region PCr-1, i.e. ". . .the BA corresponding to a return period of 50 years has not significantly decreased" does not take into account limitations in the analyses. Consequently, the discussion that follows appears to force results interpretation toward a thesis (although, I repeat, it is a thesis that I fully support). In addition, it is not clear to me if differences in other fire regime drivers such as climate, fuel flammability and landscape connectivity where considered in the model when comparing 1973-1994 and 1995-2016 periods. Indeed, if you want to test the "fire policy" driver the model should account for the variability explained by other relevant drivers. Note that in Figure 2, after 1994 the sole fire peak in the number of fires>100 ha reaching a level similar to ones before 1994 corresponds to the 2003 fire season, i.e. the major climate anomaly hitting southern France during the period of analysis. Notably, one of the author in a previous similar paper (Curt and Frejaville 2017; DOI: 10.1111/risa.12855) stresses the increase in fire weather index, human pressure and fuel coverage in the second studied period.

The authors thank the referee for these comments which will help to improve this manuscript. Please find below our answers.

In your major comments you indicate that our central argument is that the new fire policy/strategy has only reduced the return period for large fires in one pyroregion, and that this result is somewhat 'over-interpreted'. In addition, it is true that uncertainty is central in this study and these results, in our opinion, provide interesting results and clear interpretations. As in any studies with a limited availability of data (as it is the case for all geophysical studies), results must always be interpreted in light of the corresponding uncertainties. In addition to visual assessment (see, e.g. Figure 5 of the current manuscript), the Bhattacharyya

coefficient provides a quantitative assessment of the changes, taking into account the uncertainty in the estimates. This point will be included.

In addition, it is also suggested that other variables (climate, fuel flammability, and landscape connectivity) may be added to the results because they also control the area burned. In our opinion, additional analyses taking into account external drivers (other than the fire policy) are beyond the scope of this article, which aims at describing statistically the changes in extreme return levels of BA. Furthermore, these aspects have already been assessed in a previous study (Curt and Frejaville 2018). However, a discussion about the role of fire controls will be added. This will give weight to the results and reinforce our main argument which is that fire suppression policies are not sufficient, especially in pyroregions with high fire activity (PCr-1) and with increasing structural factors promoting fires.

**Minor comments:**

**Comment R2 #2.1.** Pg1, LN17, LN18 and throughout the text - Eliminate dots after "ha"

We eliminated the dots after "ha." throughout the text.

**Comment R2 #2.2.** Pg1, LN16-LN20 – report initial and end period for fire statistics listed in this paragraph

We indicated the dates of the fire data (1973-2016); the recent period corresponds to 1994-2016

**Comment R2 #2.3.** Pg2, LN1 - Include here other relevant references, e.g. Moreira et al. 2011 (DOI: 10.1016/j.jenvman.2011.06.028), Fernandes et al. 2013 (DOI: 10.1890/120298)

We included the references proposed on fire policies issues.

**Comment R2 #2.4.** Pg2, LN3-LN5 – I believe here is missing a major driver of the burnt area in southern Europe, i.e. cultural and socio-economic aspects affecting landscape management (i.e., type of urbanization, agriculture and forestry, land control, use of fire, type of postfire management) which in turn contribute to determine fire likelihood and burnt area. Note that this is supported also by authors at Pg. 3, LN15

We added references on cultural and socio-economic aspects which promote fires and burned area in the Mediterranean.

**Comment R2 #2.5.** Pg2, LN9-LN11 – While I agree knowing the return period of large fires is useful to governmental agencies and reinsurance companies to evaluate the cost of future fires, I do not believe it is useful to the dimensioning of fire crews during an extreme fire event (this is something decided in real time once the ignition point, the fire weather, potential fire trajectories and values at risk are known). Rather, as the return period of a flood is useful to the dimensioning of infrastructures such as embankments of a river (a similarity used by authors at LN 6-7), the return period of a large fire in a valley is useful to the dimensioning of fuel management measures, e.g. how many fuelbreaks, where they must be located in the landscape, how much large they must be, which is the interval between fuel treatments to maintain fuelbreaks before large fires return, which in turn determine management costs and consequently the number of fuelbreaks I can maintain in a given period.

As you indicate, knowing the return period of large fires is not of such importance for preparing/dimensioning fire crews during an extreme fire event. Our sentence was a bit misleading: this is not important during a single event (especially if it is currently ongoing), but this is important - and now better taken into account by the civil security - each year and before or during the fire seasons. Firemen account for the daily fire danger and the tendency for the weeks to come in each region in order to dimension and pre-position the fire crews on the basis of the likelihood to have a large fire in a given region. This is why this sentence will be replaced by: "Concerning fires, this information can help each year to pre-determine the size of the fire crews and of fire tactical means such as airplanes and trucks in each region, in order to support ground forces if extreme fire events occur (Lahaye et al., 2014)". We also agree that return periods of large fires can be important in order to dimension fuel management measures in a given region. It is known to be an efficient and sustainable measure for leveraging fire risk. However, as return periods of large fires are not yet calculated in France, fuel management is not currently done on this basis. We will add this comment to the text and to the discussion as a potential application of this study.

**Comment R2 #2.6.** Pg2, LN14 – later in reading the manuscript I assumed "return levels" the same as "return period", but then I realized it was not the case. However, it is not clear which is the difference between the two. Please clarify here or in the method section.

The difference between "return levels" and "return period" will be clarified in the section "Materials and Methods" of the revised paper. Return periods correspond to the average time length (e.g. 20 years, 10 years) between two return levels (100 ha, 1000 ha).

**Comment R2 #2.7.** Pg2, LN15 - after ". . .dedicated studies are available" – Although later in the paragraph authors report several references in relation to methods used to calculate the fire return period, I suggest to insert here 2-3 references to previous studies calculating the large fire return period that author think are very relevant for fire management purposes

A reference on the fire return calculation for large fires in France (Hernandez et al. 2015) will be included.

**Comment R2 #2.8.** Pg2, LN20 – after "Extreme Value Theory" add "(EVT)"

"EVT" will be added after "Extreme Value Theory".

**Comment R2 #2.9.** Pg2, LN29 – the fire policy change in 1994 in France appears here for the first time, but it is not clear in what the policy consists, and no references are provided. I would expect here, or later in the methods, a clear referring to the policy, and some quantitative data (i.e. indicators of changes in comparison to the previous policy, e.g. number of helicopters used during the fire season, annual area treated with prescribed burning) characterizing the policy. A table could be useful to synthetize information

Information on the main changes in fire policy will be incorporated. However, precise and quantitative data are often not available (e.g. the number of helicopters).

**Comment R2 #2.10.** Par 2.3 and 2.4 – Clearly state what $\mu, \sigma, \xi$ indicators means in terms of fire management

μ, σ, ξ are parameters and are not directly linked to indicators of fire management. Return levels are more easily interpreted and the focus is put on return levels rather than the GEV parameters in the remainder of the manuscript.

**Comment R2 #2.11.** Pg8, LN24 – I do not see where the "parameter uncertainty" is reported. Include model uncertainty in figure 4?

The posterior distribution is a direct assessment of the parameter uncertainty. The dispersion of the posterior distribution indicates if the parameter uncertainty is large or not. This will be clarified in the revised manuscript. Note that model uncertainty (i.e. uncertainty related to the choice of the GEV distribution) is not assessed in this study.

**Comment R2 #2.12.** Pg12, LN2 – what is meant with "median return levels"? If 20 years, change "Table 4 reports the BA corresponding to high return periods (20 and 50 years)" in "Table 4 reports the BA corresponding to median and high return periods (20 and 50 years, respectively)"

As illustrated in Figure 5, for each return period (i.e. 20 years), we can provide the whole distribution of return levels. From this distribution, we could compute any quantile, (e.g. corresponding to probabilities 0.05, 0.1, 0.9, 0.95). In Table 4, as an indicator of the central tendency of the predictive distributions of return levels, we simply report the medians, i.e. the quantile 0.5. This will be clarified in the revised manuscript.

**Comment R2 #2.13.** Figure 1 – Large fires are defined as > 1000 ha, while in the text is > 100 ha. As regards the figure caption – after "pyroclimatic regions" include "(numbered circles)", or something in the legend clarifying what colored circles represent

Thank for this comment. Fires > 1000 ha should be indicated as "very large fires". We also agree that the definition of what the colored circles represent is missing. This will be corrected in the revised manuscript.

**Comment R2 #2.14.** Figure 3 – as the aim of the paper does not focus on statistical and methodological aspects I would move figure 3 to the supplementary material

In our opinion, it is important to show statistical and methodological aspects. Figure 3 is necessary as it shows the adequacy of the GEV distributions, which is not obvious for a non-statistician. Furthermore, this comment is in contradiction with comments R1 #1.17. and R1 #1.18. made by reviewer #1 about advanced statistical aspects (distance measures, converge of the MCMC algorithm).

**Comment R2 #2.15.** Table 2 – it is not clear how it is possible to model fire return intervals > 10 years with time series of 21 years (1973-1994 and 1995-2016)

The fire return levels are obtained from the fitted GEV distributions, using Eq. (3). The GEV model is used to extrapolate beyond the time period covered by the observations.

**References**

Curt, Thomas, and Thibaut Frejaville. 2018. "Wildfire Policy in Mediterranean France: How Far Is It Efficient and Sustainable?" *Risk Analysis: An Official Publication of the Society for Risk Analysis* 38 (3): 472–88. https://doi.org/10.1111/risa.12855.